# Ligand discrimination and gating in cyclic nucleotide-gated ion channels from apo and partial agonist-bound cryo-EM structures

Jan Rheinberger[1], Xiaolong Gao[1], Philipp AM Schmidpeter[1], Crina M Nimigean[1,2,3]*

[1]Departments of Anesthesiology, Weill Cornell Medical College, New York, United States; [2]Department of Physiology and Biophysics, Weill Cornell Medical College, New York, United States; [3]Department of Biochemistry, Weill Cornell Medical College, New York, United States

**Abstract** Cyclic nucleotide-modulated channels have important roles in visual signal transduction and pacemaking. Binding of cyclic nucleotides (cAMP/cGMP) elicits diverse functional responses in different channels within the family despite their high sequence and structure homology. The molecular mechanisms responsible for ligand discrimination and gating are unknown due to lack of correspondence between structural information and functional states. Using single particle cryo-electron microscopy and single-channel recording, we assigned functional states to high-resolution structures of SthK, a prokaryotic cyclic nucleotide-gated channel. The structures for apo, cAMP-bound, and cGMP-bound SthK in lipid nanodiscs, correspond to no, moderate, and low single-channel activity, respectively, consistent with the observation that all structures are in resting, closed states. The similarity between apo and ligand-bound structures indicates that ligand-binding domains are strongly coupled to pore and SthK gates in an allosteric, concerted fashion. The different orientations of cAMP and cGMP in the 'resting' and 'activated' structures suggest a mechanism for ligand discrimination.

DOI: https://doi.org/10.7554/eLife.39775.001

*For correspondence:
crn2002@med.cornell.edu

Competing interests: The authors declare that no competing interests exist.

## Introduction

Cyclic nucleotide modulated ion channels are physiologically important for visual and olfactory signal transduction and pacemaking activity in the heart and brain (*Kaupp and Seifert, 2002*; *Craven and Zagotta, 2006*; *Robinson and Siegelbaum, 2003*; *Biel et al., 2009*). Their activity is dependent on the binding or dissociation of cyclic nucleotides (cAMP or cGMP) to a cytoplasmic cyclic nucleotide-binding domain (CNBD) within the channel. This in turn leads to changes in the ionic flow across the membrane and alterations in the membrane potential, allowing an intracellular signal such as an increase in cAMP concentration to be transmitted across the membrane of the cell. The magnitude of the ion channel response to ligand binding modulates the final cellular response and can lead to different physiological outcomes (*James and Zagotta, 2018*). Thus, it is important to understand the molecular mechanism by which binding of ligands leads to modification of the open-closed equilibrium in these ion channels.

Cyclic nucleotide-modulated channels comprise a large family of ion channels in both eukaryotes and prokaryotes (*Kaupp and Seifert, 2002*; *Craven and Zagotta, 2006*). In eukaryotes, they are categorized into two subfamilies: cyclic nucleotide-gated (CNG) and hyperpolarization-activated and cyclic nucleotide-modulated (HCN) channels. HCN channels are activated by hyperpolarizing

**eLife digest** Ion channels are essential for transmitting signals in the nervous system and brain. One large group of ion channels includes members that are activated by cyclic nucleotides, small molecules used to transmit signals within cells. These cyclic nucleotide-gated channels play an important role in regulating our ability to see and smell.

The activity of these ion channels has been studied for years, but scientists have only recently been able to look into their structure. Since structural biology methods require purified, well-behaved proteins, the members of this ion channel family selected for structural studies do not necessarily match those whose activity has been well established. There is a need for a good model that would allow both the structure and activity of a cyclic nucleotide-gated ion channel to be characterized.

The cyclic nucleotide-gated ion channel, SthK, from bacteria called *Spirochaeta thermophila,* was identified as such model because both its activity and its structure are accessible. Rheinberger et al. have used cryo electron microscopy to solve several high-resolution structures of SthK channels. In two of the structures, SthK was bound to either one of two types of activating cyclic nucleotides – cAMP or cGMP – and in another structure, no cyclic nucleotides were bound. Separately recording the activity of individual channels allowed the activity states likely to be represented by these structures to be identified.

Combining the results of the experiments revealed no activity from channels in an unbound state, low levels of activity for channels bound to cGMP, and moderate activity for channels bound to cAMP. Rheinberger et al. show that the channel, under the conditions experienced in cryo electron microscopy, is closed in all of the states studied. Unexpectedly, the binding of cyclic nucleotides produced no structural change even in the cyclic nucleotide-binding pocket of the channel, a region that was previously observed to undergo such changes when this region alone was crystallized. Rheinberger et al. deduce from this that the four subunits that make up the channel likely undergo the conformational change towards an open state all at once, rather than one by one.

The structures and the basic functional characterization of SthK channels provide a strong starting point for future research into determining the entire opening and closing cycle for a cyclic nucleotide-gated channel. Human equivalents of the channel are likely to work in similar ways. The results presented by Rheinberger et al. could therefore be built upon to help address diseases that result from deficiencies in cyclic nucleotide-gated channels, such as loss of vision due to retinal degradation (retinitis pigmentosa or progressive cone dystrophy) and achromatopsia.

DOI: https://doi.org/10.7554/eLife.39775.002

voltages and this activity is modulated by cyclic nucleotide binding (*Santoro et al., 1998*; *Lyashchenko et al., 2014*; *Wang et al., 2002*; *Zhou and Siegelbaum, 2007*; *Kusch et al., 2010*; *Ludwig et al., 1998*), while CNG channels are activated by cyclic nucleotides and their activity is only slightly modulated by depolarizing voltages (*Kaupp et al., 1989*; *Benndorf et al., 1999*; *Clayton et al., 2004*; *Nache et al., 2006*). Prokaryotic cyclic nucleotide-modulated channels, homologous in both sequence and structure to their eukaryotic counterparts, have been identified (*Nimigean et al., 2004*; *Brams et al., 2014*; *James et al., 2017*), but not investigated enough to warrant a categorization into one of these two groups.

Ligand binding and activation have been extensively studied in eukaryotic CNG and HCN channels using electrophysiology on channels expressed in heterologous expression systems (*Kaupp and Seifert, 2001*, *2002*). However, only recently, high resolution structural information for human HCN1 (*Lee and MacKinnon, 2017*) and TAX-4 (*Li et al., 2017*) (a *C. elegans* CNG channel homolog), became available with the advent of single-particle cryo-electron microscopy (cryo-EM) (*Kuhlbrandt, 2014*). For HCN1, two structures were reported, in an apo and a cAMP-bound, closed conformation, while TAX-4 is in a cGMP-bound open conformation only. HCN1 channels are activated by hyperpolarization, while the addition of saturating cAMP only marginally shifts the activation (*Santoro et al., 1998*), questioning whether cAMP is an agonist for this channel.

Different models for ligand gating have been proposed for HCN and CNG channels based on functional studies (*Biskup et al., 2007*; *Li et al., 1997*; *Ulens and Siegelbaum, 2003*). These models

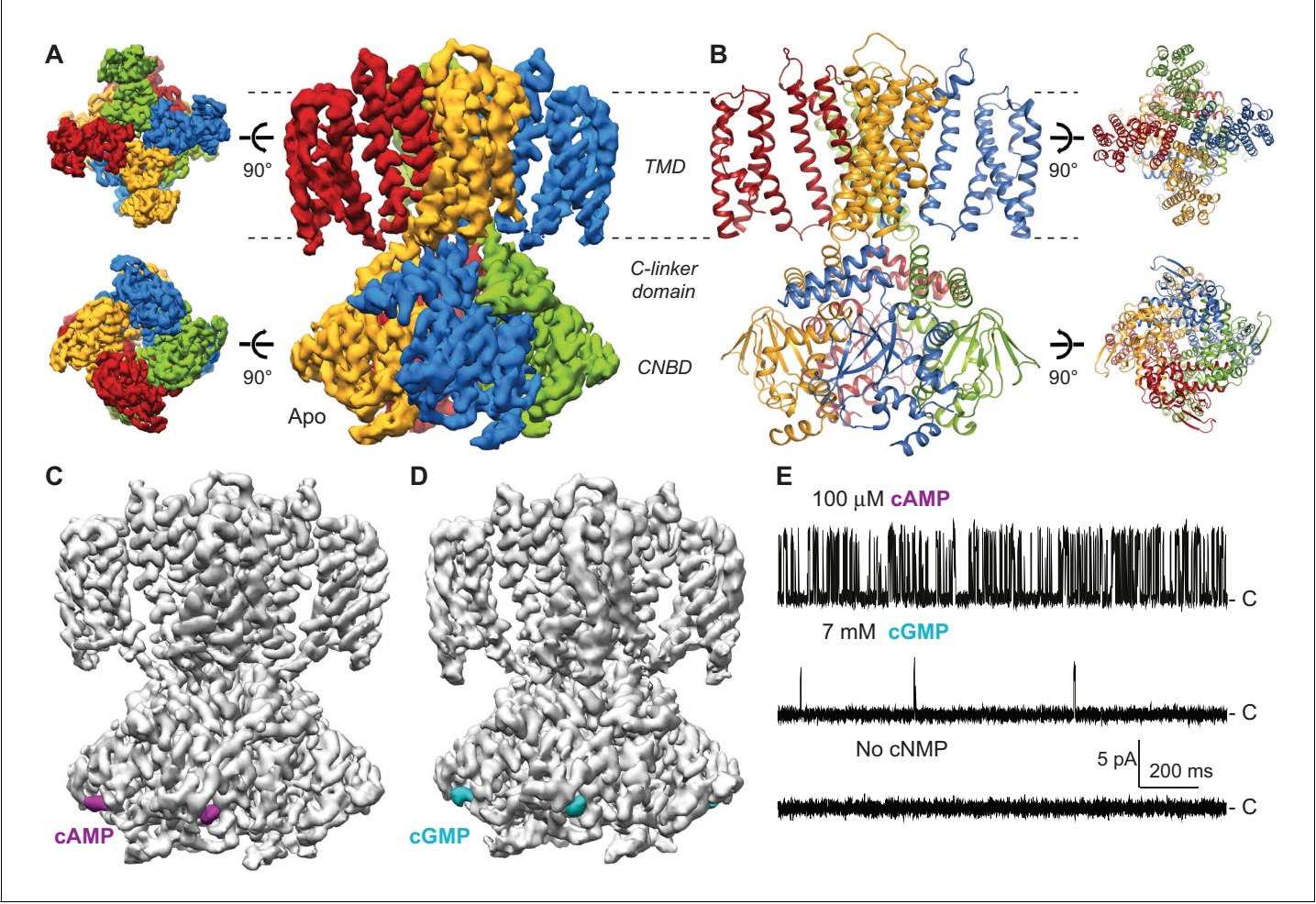

**Figure 1.** Cryo-EM structures of SthK. (**A**) Apo SthK side-view density map, colored by subunit. Dashed lines indicate the bilayer boundaries. The top is the extracellular side. Insets on the left are extracellular (top) and intracelluar (bottom) views. TMD-transmembrane domain, CNBD-cyclic nucleotide binding domain. (**B**) Atomic model of apo SthK colored as in (**A**). Insets on the right are extracelluar (top), and intracellular (bottom) views. (**C**) cAMP-bound SthK density map (grey) with bound cAMP highlighted in purple. (**D**) cGMP-bound SthK density map (grey) with bound cGMP highlighted in cyan. (**E**) Representative single-channel recording traces from SthK in horizontal lipid bilayers at +100 mV with 100 µM cAMP (top trace), 7 mM cGMP (middle), and no ligand (bottom). The zero-current level (closed channel) is indicated to the right by the letter C.

DOI: https://doi.org/10.7554/eLife.39775.003

The following figure supplements are available for figure 1:

**Figure supplement 1.** Purification of SthK in lipid nanodiscs.

DOI: https://doi.org/10.7554/eLife.39775.004

**Figure supplement 2.** Single particle cryo-EM of apo SthK.

DOI: https://doi.org/10.7554/eLife.39775.005

**Figure supplement 3.** Single particle cryo-EM of SthK in the precence of 2 mM cAMP.

DOI: https://doi.org/10.7554/eLife.39775.006

**Figure supplement 4.** Single particle cryo-EM of SthK in the precence of 7 mM cGMP.

DOI: https://doi.org/10.7554/eLife.39775.007

**Figure supplement 5.** Segmented density map of SthK-cAMP.

DOI: https://doi.org/10.7554/eLife.39775.008

**Figure supplement 6.** Sequence alignment of members from the cyclic nucleotide modulated channel family.

DOI: https://doi.org/10.7554/eLife.39775.009

are all versions or combinations of two fundamental models proposed to work in allosteric proteins: the concerted and the sequential model (*Koshland et al., 1966*; *Monod et al., 1965*). In the concerted model, all subunits of the tetrameric channel change their conformation together, all at once,

upon ligand binding and accordingly all subunits in one channel adopt the same conformation whether open or closed and irrespective of how many ligands are bound. The sequential model, in contrast, allows individual domains and/or subunits to change conformation independently upon ligand binding, one after the other, on the pathway towards the open state. The existing structures (*James et al., 2017*; *Lee and MacKinnon, 2017*; *Li et al., 2017*) are not sufficient to argue in favor of either model because there are not enough conformations available and because of the lack of direct correspondence between conformations and functional states. In order to address this void, we selected SthK, a prokaryotic cyclic nucleotide-modulated channel from *Spirochaeta thermophila* (*Brams et al., 2014*). We showed previously that SthK is activated by cyclic nucleotides and its activity is modulated by depolarizing voltages, making it more similar in this respect to CNG rather than HCN channels (*Schmidpeter et al., 2018*). The channel shares about 35% sequence similarity with eukaryotic CNG channels, and, unlike its eukaryotic counterparts, lends itself easily to high-resolution single-channel functional studies as well as structural investigations, providing a unique model to investigate ligand gating (*Schmidpeter et al., 2018*). We present here the high-resolution cryo-EM structures of SthK in the apo, cAMP-bound, and cGMP-bound states in lipid nanodiscs. This is, to our knowledge, the first apo, resting structure of a member of the CNG channel family. The single channel electrophysiology data allow us to specifically assign functional states to the SthK structures in different ligand-bound conformations, and we propose a concerted mechanism for CNG channel gating, as well as a molecular basis for cyclic nucleotide discrimination.

## Results and discussion

### SthK is a cAMP-gated potassium channel

SthK has been previously reported to function as a CNG-like channel when expressed in oocytes (*Brams et al., 2014*), and structures of the soluble ligand-binding domains in complex with both

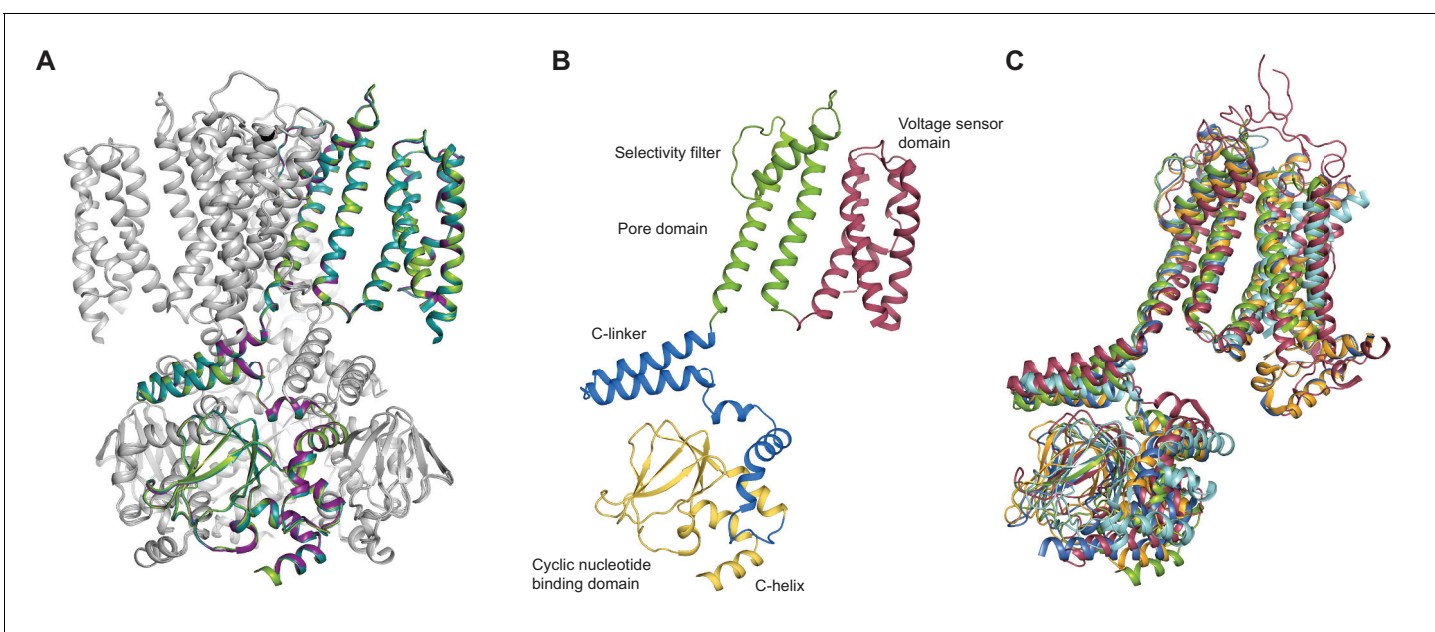

**Figure 2.** SthK channel domain architecture and structure overlays. (**A**) Overlay between the cryo-EM structures of apo (green), cAMP-bound (magenta), and cGMP-bound (cyan) SthK in cartoon representation, which shows that the three structures are nearly identical. Only one of the subunits in the tetramer is colored, the other three are in grey. (**B**) Atomic model of a single SthK subunit in cartoon representation. Red: voltage sensor, green: pore, blue: C-linker, yellow: CNBD. (**C**) Overlay between apo SthK (green), cAMP HCN1 (blue), apo HCN1 (orange), TAX-4 (red), and LliK (cyan) shows similarities in domain architecture of cyclic nucleotide-modulated ion channels. PDB codes: apo SthK$_{EM}$ (6CJQ), cAMP SthK$_{EM}$ (6CJU), cGMP SthK$_{EM}$ (6CJT), apo HCN1 (5U6O), cAMP HCN1 (5U6P), TAX-4 (5H3O), LliK (5V4S).
DOI: https://doi.org/10.7554/eLife.39775.010

cAMP and cGMP have been reported (*Kesters et al., 2015*). We recently described an SthK ion channel construct that robustly expresses in *E. coli* and we established that the purified protein is active in binding, flux, and single-channel electrophysiology assays (*Schmidpeter et al., 2018*). Purified and reconstituted SthK showed no activity in cyclic nucleotide-free conditions, became active upon cAMP application, channel activity was increased upon membrane depolarization and the channel showed no evidence of inactivation (*Schmidpeter et al., 2018*) (*Figure 1*, Figure 4B). Nevertheless, the maximal open probability of SthK in saturating concentrations of cAMP was less than 1, even in the presence of depolarizing voltages, indicating that cAMP is a partial agonist for SthK. The cAMP-induced activity is inhibited by cGMP application, although infrequent single-channel openings are still observed (*Schmidpeter et al., 2018*). Unlike in cyclic nucleotide-free conditions, the infrequent openings are still present in high concentrations of cGMP alone, indicating that cGMP is not an antagonist for SthK, as has been previously proposed based on patch-clamp recordings from oocytes expressing SthK (*Brams et al., 2014*), but rather an extremely poor partial agonist (*Figure 1E*). The difference in attribution is likely due to the extremely low open probability of SthK in cGMP making it easy to miss when recording from cells, or from the difference in the experimental setup (purified channels in controlled lipid bilayer composition versus channels expressed in oocytes).

In order to get insights into partial agonism in SthK, we solved the structures of SthK with single-particle cryo-EM in three different states: apo (cyclic nucleotide-free conformation), cAMP-bound (partial agonist-bound conformation), and cGMP-bound (poor partial agonist-bound conformation) (*Figure 1A, C and D*). The three structures are with the SthK channel reconstituted in lipid nanodiscs, in order to recapitulate a native-like membrane environment (*Figure 1—figure supplement 1*). The structures were solved to similar resolutions, 3.42, 3.35, and 3.46 Å for the apo, the cAMP-bound, and the cGMP-bound states, respectively (*Figure 1—figure supplements 2–5*, *Supplementary file 1*).

The apo structure adopts an architecture different from any of the existing CNG/HCN structures (*Lee and MacKinnon, 2017*; *Li et al., 2017*), which is expected, as it is the first apo, resting state structure of a member of the CNG channel family (*Figure 2*). Unexpectedly however, the cAMP- and cGMP-bound SthK structures also display a resting, apo-like ligand binding pocket, despite the resolved, bound ligands (*Figures 1* and *2*). Thus, the CNBDs of the ligand-bound full-length SthK adopt a different conformation than that observed in the crystal structures of the isolated ligand-bound CNBDs (*Figures 1* and *2* and *Figure 6—figure supplement 1*). Since the three SthK structures have largely similar features, and alignments between them yield RMSDs that are very small (apo/cAMP: 0.50 Å, apo/cGMP: 0.40 Å, cAMP/cGMP: 0.46 Å), we will focus first on the apo structure, and then discuss the unanticipated effects of ligand binding and their implication for mechanism.

## SthK architecture

Similar to the eukaryotic cyclic nucleotide-modulated ion channels whose structures were solved recently by cryo-EM (HCN1, TAX-4) (*Lee and MacKinnon, 2017*; *Li et al., 2017*), the SthK channel is a functional tetramer and the structure can be divided in three layers as viewed from the membrane plane (*Figure 1A*). The top layer contains the transmembrane domain (TMD) with the ion-conducting pore, and a voltage sensor-like four transmembrane helix bundle, which presumably is responsible for voltage modulation of the channel (*Figures 1* and *4*). The voltage sensor-like domains are in a non-swapped configuration, similar to HCN1, TAX-4, and LliK (*James et al., 2017*), but unlike MloK1 (*Kowal et al., 2018*). The bottom layer, at the intracellular side, consists of four cyclic nucleotide binding domains (CNBDs), which contain the ligand-binding pockets (*Figure 1A*). Sandwiched in between the top and bottom layers is the C-linker domain, strategically located to relay the ligand binding event from the CNBDs to the channel pore (*Figure 2B*). The C-linker/CNBD complex is domain-swapped with respect to the transmembrane region (*Figures 1A* and *2A*). Some prokaryotic homologs of cyclic nucleotide-modulated channels with structures solved by cryo-EM completely lack a C-linker (such as MloK1) (*Kowal et al., 2018*; *Chiu et al., 2007*), while others have an intact C-linker (such as SthK and LliK) (*James et al., 2017*; *Kesters et al., 2015*). Out of these prokaryotic channels, only SthK is conducive to being analyzed in detail at the functional level with single-channel recordings. Sequence alignments as well as structure overlays of SthK with HCN1, TAX4, and

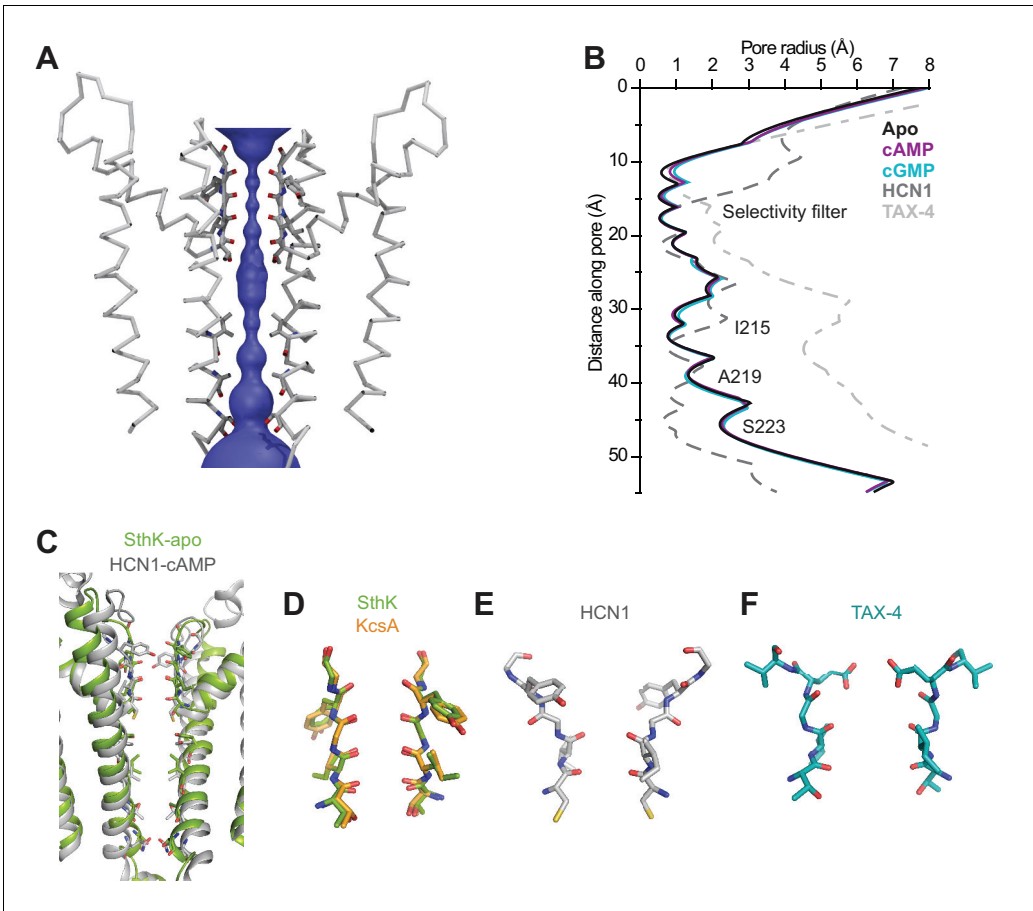

**Figure 3.** SthK channel pore and selectivity filter. (**A**) SthK pore domain with selectivity filter carbonyls highlighted in red. Only two opposing subunits are shown to highlight the size of the water-filled pore, purple. (**B**) Pore radius as a function of the distance along the pore calculated with the program HOLE for all three SthK structures and compared to those for HCN1 and TAX-4. Lines colors and types as indicated in the figure. (**C**) Overlay of SthK (green) and HCN1 (grey) pore region. Side chains facing towards the pore axis are displayed as sticks. Selectivity filter of SthK-KcsA overlay (**D**), HCN1 (**E**) and TAX-4 (**F**) in stick representations.

DOI: https://doi.org/10.7554/eLife.39775.011

LliK, show that all channels are similar in coarse overall architecture (*Figure 1—figure supplement 6* and *Figure 2C*).

## Ion channel pore

SthK is a potassium selective channel (*Brams et al., 2014*), and displays a traditional potassium channel pore architecture (*Zhou et al., 2001*; *McCoy and Nimigean, 2012*) where the selectivity filter, encoded by the TIGYGD signature sequence for potassium selection (*Heginbotham et al., 1992*, *1994*) (*Figure 1—figure supplement 6*), is the same as that of KcsA and other potassium channels (*Figure 3*). The pore widens below the selectivity filter into an aqueous cavity. Roughly half way down the membrane, in the middle of this cavity, the side chains of I215 pointing radially towards the pore axis, constrict the pore to a diameter of the size of a dehydrated potassium ion similar to what is seen in the HCN1 and LliK channels pore (*James et al., 2017*; *Lee and MacKinnon, 2017*) (*Figure 3A–B*). However, instead of further narrowing into a bundle-crossing, like in HCN1 or KcsA (*Lee and MacKinnon, 2017*; *Liu et al., 2001*), the SthK pore widens towards the intracellular side. The expectation from the functional data (*Schmidpeter et al., 2018*) is that the SthK pore in the cryo-EM structures is in a closed conformation as will be discussed in further detail (*Figure 1E*, *Figure 4—figure supplement 2*). Thus, it is possible that the gate in SthK is located at a previously

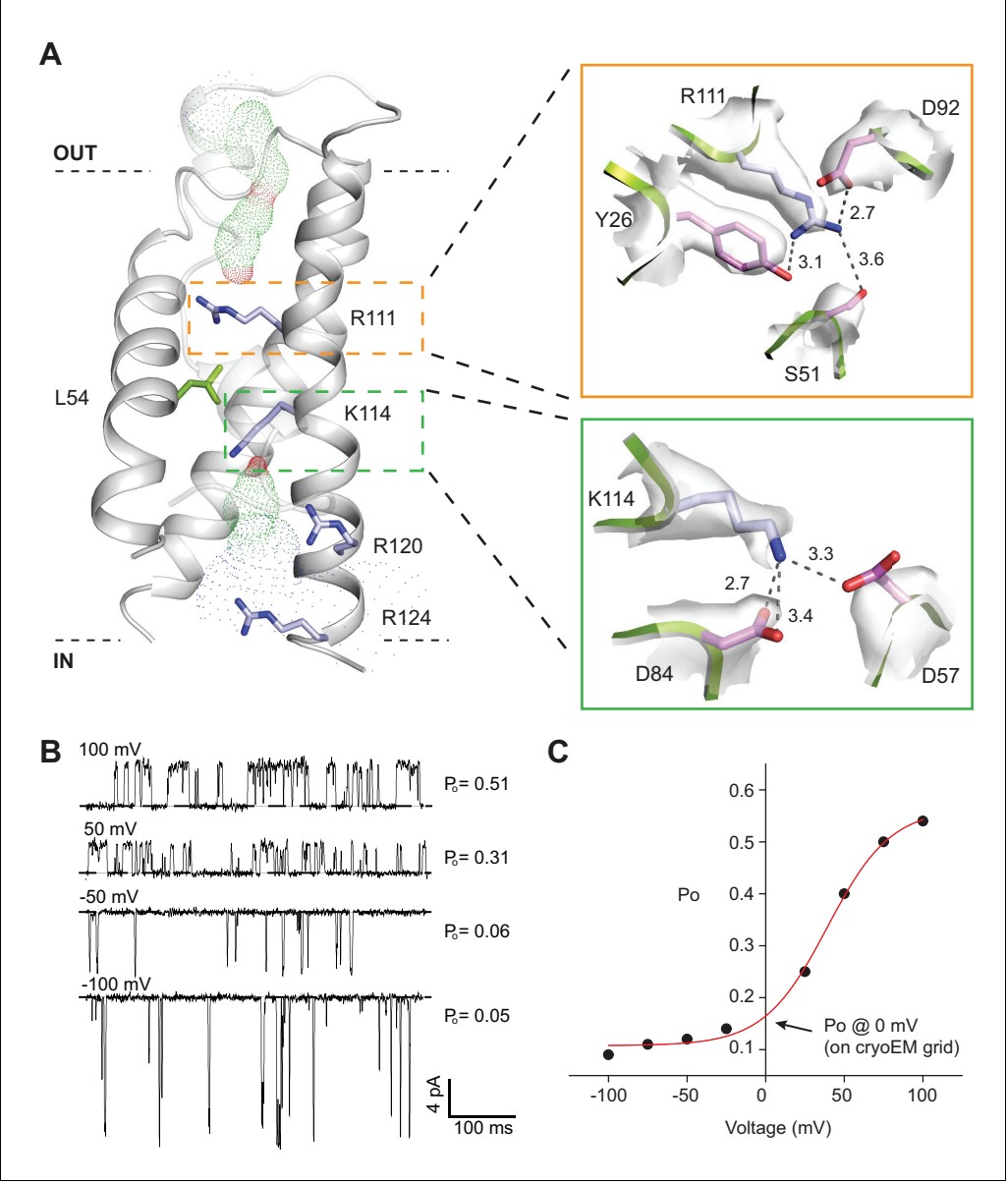

**Figure 4.** SthK channel voltage dependence. (**A**) SthK voltage sensor domain (grey) with the 4 positively-charged residues in S4 in blue and L54 in S2 in green. The boxes zoom into the coordination between R111 and countercharges in S3 and S2 (orange) and between K114 and countercharges in S3 and S2 (green). Experimental density shown in light grey at 6 σ (top) and 4 σ (bottom). Dot representation displays water accessible cavities in the voltage sensor calculated with the program HOLE (red: pore radius <1.15 Å; green: pore radius 1.15–2.30 Å; blue: pore radius >2.30 Å). (**B**) Representative single-channel recording traces from SthK at different voltages (indicated, top of traces) displaying different open probabilities (indicated, right). The dashed line indicates the closed channel level. The traces were filtered online at 1 kHz and additionally filtered offline at 500 Hz. The ligand concentration was 1 mM cAMP. (**C**) Open probability as a function of voltage for the SthK channel for which selected traces are displayed in (**B**). Arrow indicates the open probability value at 0 mV, which is the voltage the channels experience on the cryo-EM grid. This experiment was performed for at least 7 different bilayers containing single SthK channels in saturating cAMP concentrations, and a plot showing an average Po vs. voltage curve is shown in ref. *Schmidpeter et al. (2018)*.

DOI: https://doi.org/10.7554/eLife.39775.012

The following figure supplements are available for figure 4:

**Figure supplement 1.** Voltage sensor domains in SthK and related channels.

DOI: https://doi.org/10.7554/eLife.39775.013

*Figure 4 continued*

**Figure supplement 2.** SthK single-channel analysis.

DOI: https://doi.org/10.7554/eLife.39775.014

unreported location, in the middle of the cavity, which is the second narrowest point in the permeation pathway, after the selectivity filter. However, we cannot exclude the possibility that an additional gate exists at the level of the selectivity filter, similar to other ligand-gated ion channels (*Heer et al., 2017*; *Posson et al., 2015*; *Wilkens and Aldrich, 2006*), including eukaryotic CNG channels (*Flynn and Zagotta, 2001*).

## Voltage sensor domain in the resting state

SthK is activated by cAMP binding and this activity is further increased with depolarization (*Schmidpeter et al., 2018*) (*Figure 4B and C*) similar to other voltage-gated $K^+$ and $Na^+$ channels (*Bezanilla, 2000*), but unlike HCN channels that activate with hyperpolarization (*Santoro et al., 1998*; *Wang et al., 2002*). The channel has a voltage sensor-like domain S1-S4, and the S4 transmembrane helix contains 4 basic residues (*Figure 4A*, *Figure 4—figure supplement 1*). Although we do not have direct evidence that this is a *bona fide* voltage sensor, the similarities to other known voltage sensor domains and the increase in channel activity with depolarization lead us to hypothesize that the S1-S4 helices in SthK also form a voltage sensor. Two out of the four basic residues in

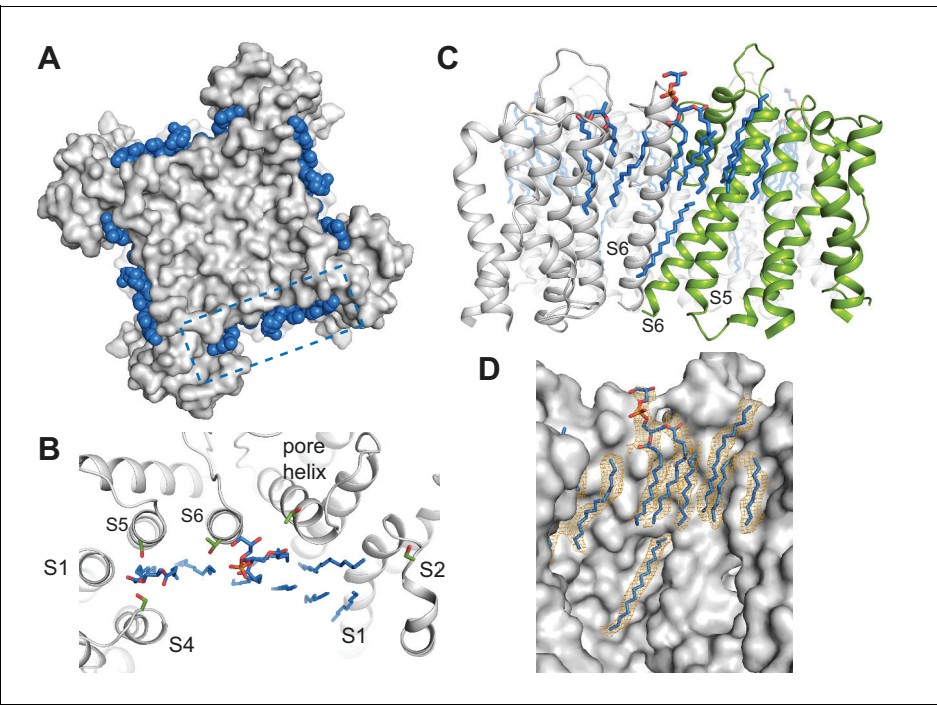

**Figure 5.** Specific lipid-protein interactions in SthK. (A) View from the extracellular side showing a surface representation of SthK (grey) with the annular lipids (blue) located along the protein periphery. Dashed blue box indicates the region shown in B. (B) Detailed representation of location and possible interactions of the lipids (blue) with amino acid side chains (green) along the extracellular side. Transmembrane helices (grey) are labeled as S1-S6. (C) Side view of the transmembrane domain with all resolved lipids. One subunit was colored green for better visualization. (D) Same view, zoomed-in, of the transmembrane domain, showing the lipid density (in orange mesh) and the modeled lipids (blue). Protein surface representation (grey). Densities shown are at a contour level of 5 σ.

DOI: https://doi.org/10.7554/eLife.39775.015

S4, R111 and K114, are located within the electric field drop across membrane and are likely to contribute to the gating charge, while R120 and R124 are solvent exposed due to a cavity on the intracellular side of the S4 helix (*Figure 4A*). The small number of arginine and lysine residues in the S4 helix of SthK is in agreement with the lower voltage dependence of this channel compared with other voltage-gated channels, which have up to 9 positively-charged residues in the membrane (*Bezanilla, 2000*; *Tao et al., 2010*). L54 on the S2 helix, at the equivalent position of the phenylalanine, the so-called charge transfer center in voltage-gated K⁺ channels (*Tao et al., 2010*) (*Figure 1—figure supplement 6*), has its side chain protruding right in between R111 and K114 (*Figure 4A*), suggesting that L54 is the charge transfer center in SthK. Although most charge transfer centers are either phenylalanines or tyrosines, the voltage-gated KvAP potassium channel also has a leucine at this position, similar with SthK (*Jiang et al., 2003*) (*Figure 4—figure supplement 1*). Since the channel experiences a membrane potential of 0 mV on the EM grid, and the median value for voltage activation is about 85 mV (*Schmidpeter et al., 2018*), the voltage sensor is likely in the down, resting conformation in the SthK structure (*Schmidpeter et al., 2018*) (*Figure 4B–C*). Furthermore, both R111 and K114 appear strongly anchored in place by multiple attractive interactions within short distances with residues from both S2 and S3 helices (*Figure 4A*), in agreement with the requirement for strong depolarizations needed to activate the channel. In comparison, the positively-charged residues on the S4 helix of HCN1 (easier to open with voltage than SthK) appear less

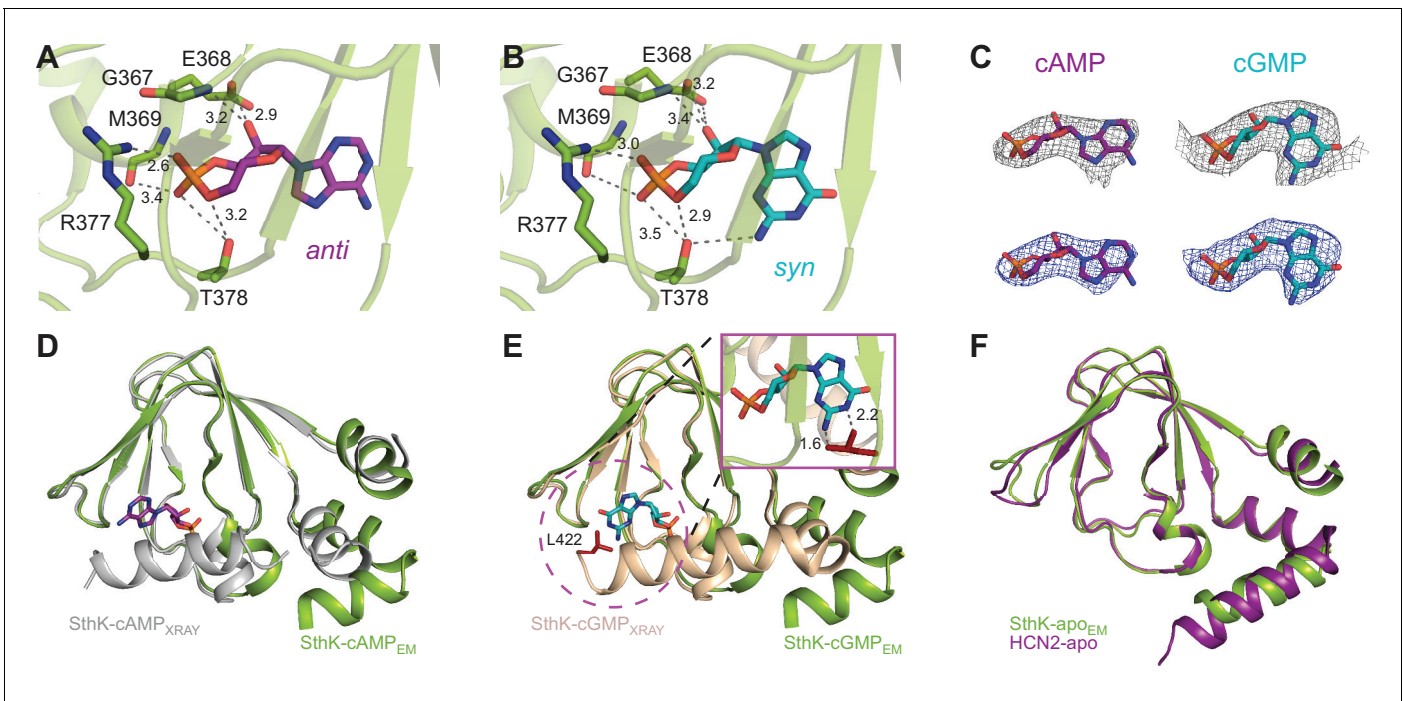

**Figure 6.** Ligand binding in the SthK CNBDs. (**A**) cAMP bound in anti configuration to the binding pocket. (**B**) cGMP bound in the syn configuration having similar interaction with SthK. (**C**) Density maps (grey mesh) for cAMP (purple, anti) and cGMP (cyan, syn) showing the ligand fit. The corresponding difference maps (blue mesh) between ligand-bound and apo experimental density maps confirmed ligand binding. Densities shown are at a contour level of 5.5 σ. (**D**) Overlay between the cAMP-bound SthK CNBD from cryo-EM (green) and from crystallography (grey, PDB:4D7T). (**E**) Overlay between cGMP-bound SthK CNBD from cryo-EM (green) and from crystallography (wheat, PDB:4D7S). Insert shows potential clash of the C-helix (L422, highlighted in red) from the crystal structure with the cGMP in the syn configuration. (**F**) Overlay between apo SthK (green) and apo HCN2 (purple) CNBDs showing identical conformations.

DOI: https://doi.org/10.7554/eLife.39775.016

The following figure supplements are available for figure 6:

**Figure supplement 1.** Comparisons between CNBDs of SthK and related channels.
DOI: https://doi.org/10.7554/eLife.39775.017

**Figure supplement 2.** Difference density maps for cyclic nucleotides in the CNBDs of SthK.
DOI: https://doi.org/10.7554/eLife.39775.018

strongly anchored in place with counter charges than those in SthK, while those on TAX-4 (quasi voltage-insensitive) appear more strongly anchored (*Figure 1—figure supplement 6*, *Figure 4—figure supplement 1*). Thus, there is a correlation between the strength of the interactions the voltage-sensitive charges in S4 make with their counter charges in S2 and S3 and the energy necessary to stimulate the channels with voltage.

## Lipid interactions

All three SthK structures revealed multiple extraneous elongated densities running along and in direct contact with the transmembrane helices, at interfaces between helices, and in crevices along subunit interfaces (*Figure 5*). Since these channels are in lipid nanodiscs, and we see these densities in each of the three structures at the same positions, we assigned them to lipid molecules binding specifically to SthK (POPG, since the nanodiscs were assembled with this lipid only). In all cases, only partial lipid molecules were modeled, since the densities did not support modelling of most headgroups or the entire length of the alkyl chains. The lipid molecules modeled are all annular lipids and most of them are observed to partition in the top leaflet of the bilayer (*Figure 5*), suggesting that the top transmembrane part of the channel is strongly anchored in the membrane. The lipids also bind at the interfaces between adjacent subunits and in the crevices between the voltage sensors and pore domains, perhaps acting as a greasy slide during voltage gating, consistent with our previous finding that SthK gating is dependent on the lipid composition of the bilayer (*Schmidpeter et al., 2018*).

The dearth of observed lipids on the inner leaflet is consistent with the idea that the inner helices are moving during gating in CNG channels (*Flynn and Zagotta, 2001*). We identified only one lipid molecule at the inner leaflet of the bilayer, bound along the length of the S6 helix all the way up to the end of the pore helix/beginning of the selectivity filter (*Figure 5C–D*). Its unusual position near the moving channel parts that form the gates is also consistent with the role of lipids in SthK gating (*Schmidpeter et al., 2018*). The lipid-SthK interactions are mostly hydrophobic, although multiple serines and threonines were found near the putative locations of the headgroups at the extracellular side suggesting that some specific interactions exist. Similarly located lipids have been identified in the structure of Kv1.2 channels crystallized in the presence of lipids (*Long et al., 2007*), and in the cryo-EM structure of other channels in nanodiscs (*Gao et al., 2016*; *Autzen et al., 2018*; *Jin et al., 2017*; *Dang et al., 2017*; *Chen et al., 2017*).

## C-linker/CNBD

The C-linker/CNBD domain of SthK is overall highly homologous in both sequence and architecture/structure with the C-linker/CNBD of all other cyclic nucleotide-modulated channels (*Figure 1—figure supplement 6*, *Figure 6—figure supplement 1*), with an important difference. Previous crystal structures of isolated C-linker/CNBD domains of other CNG/HCN channels, and, importantly, also of SthK itself, allowing direct comparisons, have the most C-terminal helix of the CNBD (the C-helix) packed tightly like a lid against the binding pocket and interacting closely with the ligand (*Figure 6D–E*, grey and wheat, *Figure 6—figure supplement 1*) (*Lee and MacKinnon, 2017*; *Li et al., 2017*; *Kesters et al., 2015*; *Saponaro et al., 2014*). We will call this the 'activated' CNBD conformation. In all our full-length SthK structures, whether apo, or ligand-bound, the C-helix is swung away from the binding pocket (*Figure 6D–E* green), most similar to the only reported apo CNBD conformation of HCN2 (*Goldschen-Ohm et al., 2016*) (*Figure 6F*). We will call this the 'resting' CNBD conformation. Furthermore, due to the lower positioning of the C-helix in our full-length SthK structures, the loop that connects the C-linker with the CNBD (which we previously called the siphon loop [*Kowal et al., 2018*]) also has more room to expand downward further than in the 'activated' CNBDs (*Figure 6—figure supplement 1*). Interestingly, not only our apo, but also the cAMP- and cGMP-bound full-length SthK structures display the CNBD in a 'resting' conformation. The presence of the ligand in the ligand-binding pocket of our full-length SthK structures is shown as experimental maps (grey mesh in *Figure 6C*) as well as difference maps between ligand-bound and apo density maps (blue mesh *Figure 6C*). The entire ligand density is recovered in the difference maps indicating that there is no density in the binding pocket in the experimental apo map (compare blue and grey mesh in *Figure 6C*).

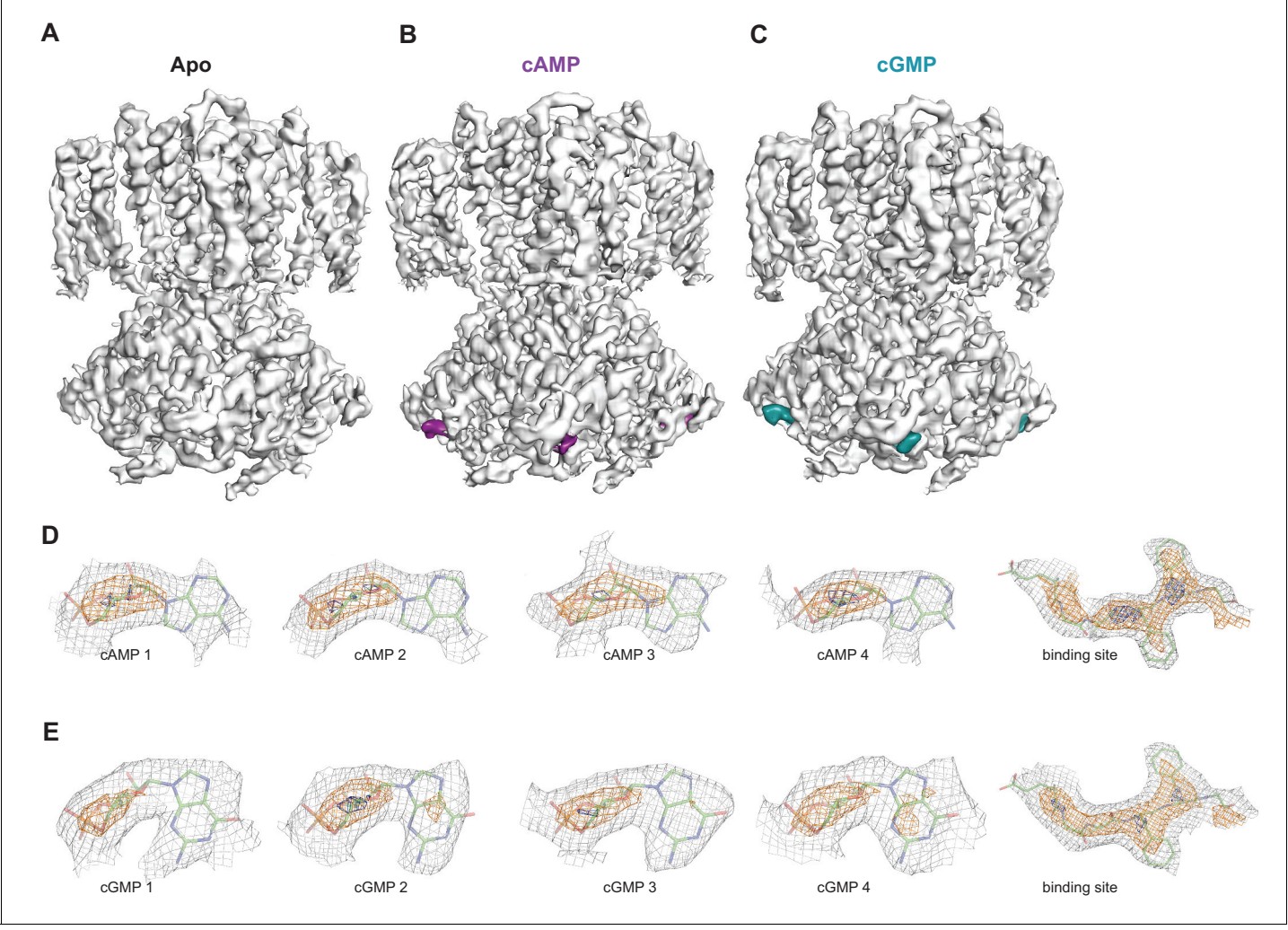

**Figure 7.** Ligand density in non-symmetrized SthK maps. Full-length, non-symmetrized C1 density maps for apo (**A**), cAMP-bound (**B**), and cGMP-bound (**C**) SthK. Overall protein density is in grey surface, cAMP density is in magenta (**B**), and cGMP density is in cyan (**C**). cAMP (**D**) and cGMP (**E**) density for each binding site in the four subunits of the tetramer, contoured at 6 σ (grey mesh), 11 σ (orange mesh), and 15 σ (dark blue mesh). All four ligand densities (for cAMP and cGMP) decrease at the same rate as the contour levels increase, and disappear at ~15 σ, and at the same rate as the protein density in a nearby region (rightmost panels in **D**, **E**), suggesting that all binding sites are occupied with ligand and the occupancy is close to 1 in each of the sites. Ligands and protein are drawn in stick representation.

DOI: https://doi.org/10.7554/eLife.39775.019

In order to investigate whether each individual CNBD within a tetramer is actually occupied by ligand, we also performed 3D refinement without imposing C4 symmetry on the ligand-bound particles (C1, unsymmetrized maps for apo, cAMP-bound and cGMP-bound SthK are shown in *Figure 7A–C*). Although at a lower resolution (3.6 Å, see *Supplementary file 1*), the final refined structures were nearly identical to the fourfold symmetrized structures we present here, with minor differences between the subunits within the tetramer, suggesting that all subunits within the tetramer are the same. The ligand is bound with an occupancy close to 1 in each of the 4 subunits of the tetramer for both cAMP (*Figure 7D*) and cGMP (*Figure 7E*), suggesting that all four subunits are ligand-bound.

We modeled cAMP to bind to the CNBD in an *anti*-configuration (*Figure 6A*), although the density can also accommodate the *syn*, albeit not as well (*Figure 6—figure supplement 2*). On the other hand, cGMP can be modeled much better in the *syn* configuration than in *anti* in the

experimental map, likely due to the favorable interaction between the hydroxyl group of T378 and the 2-amino group of the guanosine (*Figure 6B–C*, *Figure 6—figure supplement 1*), which cannot occur if cGMP is in *anti* (*Kesters et al., 2015*). The equivalent threonine in other CNG channels has been previously shown to play a role in the cGMP binding (*Altenhofen et al., 1991*). The other interactions between the ligands and the binding pocket are conserved (*Figure 6A–B*), and have been described before in detail (*Kesters et al., 2015*).

Given its accommodating density, the lack of interaction between the base of the cAMP molecule and the protein binding pocket, as well as the 70–30% solution distribution of the *anti-syn* conformers of cAMP, respectively (*Yathindra and Sundaralingam, 1974*), it makes sense that both conformers would bind to the protein, likely in a similar distribution as seen in solution. For cGMP on the other hand, the fact that the experimental map accommodates the *syn* conformer much better, only the *syn* conformer can make a favorable interaction with the protein, and the solution distribution of the *syn-anti* cGMP is 95–5%, respectively (*Yathindra and Sundaralingam, 1974*), strongly indicates that cGMP binds in *syn* to SthK. The somewhat featureless aspect of the experimental cAMP map despite its higher resolution (cAMP-bound SthK is 3.3 Å, while cGMP-bound SthK is 3.5 Å resolution, *Supplementary file 1*) also argues towards it representing a mixture of the two cAMP configurations, although we cannot rule out that the features distinguishing *syn* from *anti* cAMP are absent due to a low overall definition of the cAMP in the map.

## Partial agonist effect of cAMP binding reflected in the structure

As described above, the full-length channel, including the ligand binding pocket, adopts the same resting/closed conformation not only in the apo but also in the cAMP-bound state. We used single-channel analysis to define the functional state of SthK (open/closed) in the conditions employed to determine the structure with cryo-EM. In saturating cAMP concentrations, the open probability (Po) of SthK is low (*Schmidpeter et al., 2018*) (~0.2–0.5, at +100 mV, *Figures 1* and *4*), indicating that cAMP is only a partial agonist and not a full agonist for SthK. Additionally, as we showed before (*Figure 4B,C*), SthK activity is increased by depolarization, and thus in the conditions on the microscope grid (at 0 mV, where the voltage sensor is not activated yet) the open probability is even lower (~0.05–0.1) and the open intervals are very short (*Schmidpeter et al., 2018*) (*Figure 4*, *Figure 4—figure supplement 2*). Thus, we expect that at most 5–10% of the channels imaged from a grid frozen in saturating cAMP conditions are in the open state. Moreover, despite our efforts towards detecting the open state, 3D classification never produced a class that could correspond to an open state, thus the channels that contribute to the final density map are mostly in a closed state.

The potential mixed-configuration binding of the cAMP in the binding pocket may provide an explanation for its partial agonist nature. As discussed above, cAMP may bind to apo SthK in both the *syn* and the *anti* configurations (*Figure 6A,C*, *Figure 6—figure supplement 2*). However, the 'activated' SthK CNBD (*Kesters et al., 2015*) displays cAMP bound in the *anti* configuration (*Figure 6D*). Clashes are observed if we artificially dock *syn* cyclic nucleotides in the binding pocket of the 'activated' SthK CNBD suggesting that only *anti* cAMP binding can lead to activation (*Figure 6D–E*, insert, shown for cGMP). Thus, only a subset of all cAMP binding events (those where cAMP is in *anti*) can lead to channel opening, likely contributing to the subunitary open probability of SthK in saturating cAMP. On the other hand, in HCN2 channels, where cAMP is an efficacious agonist (*Wainger et al., 2001*; *Zagotta et al., 2003*), *syn* cAMP may not clash with the C-helix of HCN2 channels and may even make favorable interactions due to the different amino acid composition of the HCN2 C-helix (*Figure 1—figure supplement 6*).

## Weak-partial-agonist effect of cGMP binding reflected in the structure

The cGMP-bound SthK structure is closed, in agreement with the role of cGMP as very poor partial agonist (Po ~0.003, *Figure 1E*), where less than 0.3% of the channels on the cryo-EM grid are expected to be in the open state. SthK discriminates strongly between cAMP and cGMP in the degree with which each of these ligands promote channel opening. The weak activation of SthK by cGMP may be explained by the different binding orientations of cGMP between the resting and 'activated' CNBDs. As we pointed out above, cGMP can be better accommodated in the *syn* configuration in the density map (*Figure 6B,C,E*, *Figure 6—figure supplement 2*). This was surprising,

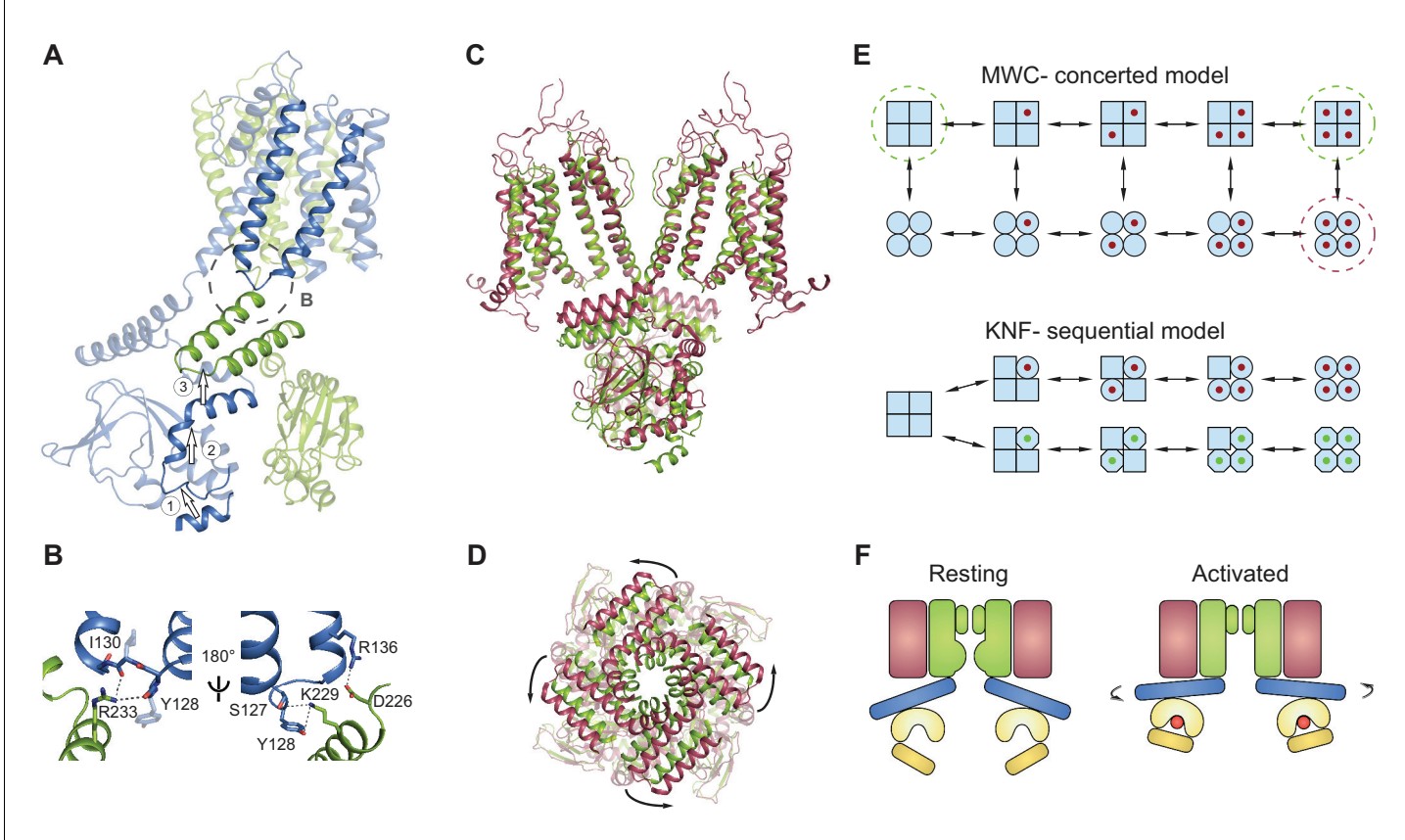

**Figure 8.** Gating model for SthK channels. (A) Two adjacent subunits of SthK (green and blue) illustrating the molecular motions needed to transition from the resting to the 'activated' CNBD conformation seen in the crystal structure. The numbers and arrows are used to describe the motions and directionality. (B) Specific interactions between the C-linker and the S4-S5-linker from the adjacent subunit indicated in the dashed circle in A. Left: same orientation as in A; right: rotated by 180°. Potential interactions between protein regions are shown with dotted lines. (C) Overlay between apo SthK (green) and the open TAX-4 (red) structures showing the upward displacement of the C-linker and the CNBD (especially the C-helix) during channel activation. Only two opposite subunits are shown for clarity. (D) The overlay of the same two structures viewed from the extracellular side showing only the C-linker/CNBD and two helical turns of S6 reveals an additional counterclockwise rotation of the C-linker that pulls on the ends of the S6 helices resulting in enlargement of the base of the pore. (E) Cartoon of two gating models. Blue squares represent the resting and blue circles the activated states. The MWC-model (top) proposes concerted conformational changes from closed to open states (all squares become all circles). The red dot is the ligand. Dashed green circles highlight the resting, closed SthK states solved with cryo-EM and presented here. Dashed red circle indicates the open, activated TAX-4-like state. The KNF model (bottom) proposes sequential changes in individual subunits as ligands bind on the path towards open states (one square becomes a circle when red ligand binds and an octagon when green ligand binds). (F) Cartoon of activation depicting domain rearrangements from the resting/closed state (left) to the activated/open state (right). Colors as in *Figure 2B*. Curved arrows indicate the C-linker rotation shown in D.

DOI: https://doi.org/10.7554/eLife.39775.020

given that cGMP was found to bind in an *anti* configuration in the 'activated' CNBD of SthK (*Kesters et al., 2015*) (*Figure 6—figure supplement 1*). The binding pocket (*Figure 6B*) likely prefers the *syn* cGMP due to the additional interaction with T378 also mentioned above, which cannot occur when the cGMP is in *anti*, and also because of the *syn*-biased distribution of cGMP configurations in solution. On the other hand, *syn* cGMP cannot be easily accommodated in the 'activated' CNBD binding pocket, likely due to a clash with the side chain of L422 on the C-terminal C-helix (*Figure 6E*, insert). We propose that the reason for the weak partial agonism of cGMP for SthK is that the apo conformation strongly prefers to bind the *syn* cGMP, which does not allow the CNBD to 'activate'.

This is in contrast to TAX-4 and the isolated CNBD from HCN2, where *syn* cGMP stabilizes the 'activated' CNBD via favorable interactions between the base and the C-helix, which are absent in

SthK (*Li et al., 2017*; *Zagotta et al., 2003*). In these channels, cGMP is able to increase activity to a large degree, so it makes sense that the form of cGMP that binds with higher affinity to the apo conformation is able to support the conformational change.

## SthK channels gate in a concerted manner

Surprisingly, the binding of either cAMP or cGMP to full-length SthK does not induce the local conformational changes in the binding pockets that have been observed in the crystal structures of ligand-bound isolated SthK CNBDs (*Kesters et al., 2015*) (*Figure 6D–E*). This can occur if strong allosteric coupling exists between the CNBD and the pore in the full-length channel. We propose that when ligand binds to the isolated CNBD, the binding energy is sufficient to activate it and change its conformation, but it is not enough to activate the CNBD when it is part of the full-length channel, suggesting that the CNBD and the pore domain must change conformation together, in a concerted way.

The absence of a local conformational change in the binding pocket despite the bound ligand can thus be understood in the context of a concerted Monod-Wyman-Changeux (*Monod et al., 1965*) (MWC) model, previously proposed to describe the gating of CNG/HCN channels and other multi-subunit ligand-gated ion channels (*Li et al., 1997*; *Colquhoun and Sivilotti, 2004*; *Lape et al., 2008*). This model stipulates that the channel exists in two conformations: closed (or resting), and open (or activated) (*Figure 8E*, top). All closed conformations, whether apo or ligand bound, are identical, differing only in the number of ligands bound (all squares in *Figure 8E*, top). Each closed channel can undergo a concerted conformational change to an open channel, meaning that all subunits undergo the closed to open transition at the same time. All open conformations are also identical to each other (all circles in *Figure 8E*, top). This model requires that the difference between partial agonists and full agonists lies only in the efficacy with which the agonist can induce the concerted conformational change (the vertical transition from all squares to all circles in *Figure 8E*, top). The conformational change is favored when efficacious agonists are bound, so that most bound channels are open. Conversely, the conformational change is less favored when less efficacious agonists are bound, so that most bound channels are closed. Our data shows identical, closed structures for the apo and inefficacious, partial agonist-bound SthK channels, which is consistent with the concerted gating model (the functional states - closed - that we solved with cryo-EM are indicated by dashed green circles in *Figure 8E*, top). In contrast, identical structures are not expected from ion channels gating according to the sequential (or Koshland-Nemethy-Filmer, KNF [*Koshland et al., 1966*]) model, which predicts that binding of partial agonists and full agonists will induce specific closed conformational states (the red and green ligands leading to circles and octagons, respectively in *Figure 8E*, bottom), different than the apo state (squares) and perhaps different from each other (circles versus octagons in *Figure 8E*), leading to channel opening. Further ruling out the sequential model, we found no major differences between the subunits within the tetramer when the structural data was processed without fourfold symmetrization (*Figure 7A–C*) and all ligands were bound to an occupancy close to 1 in each of the subunits within the tetramer (*Figure 7D–E*).

Previous functional investigations employed concerted, sequential, or a combination of both models to describe gating in CNG and HCN channels (*Biskup et al., 2007*; *Li et al., 1997*; *Ulens and Siegelbaum, 2003*). To our knowledge, this study provides the first structural evidence that a concerted step exists in the gating of CNG channels. Conversely, other ligand-gated channels, such as the AMPA receptors, have been proposed to gate according to the sequential model, based on findings that the structures of the ligand binding domains display major differences depending on the type of agonist/antagonist bound (*Armstrong and Gouaux, 2000*; *Armstrong et al., 2003*; *Jin et al., 2003*; *Sobolevsky et al., 2009*; *Twomey et al., 2017*).

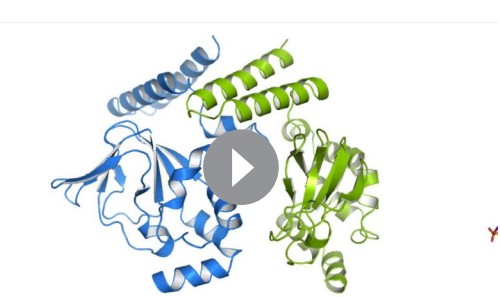

**Video 1.** Agonist-induced movement of the C-linker/CNBD of SthK by interpolating the structures of SthK apo with that of the isolated SthK CNBD crystal structure. DOI: https://doi.org/10.7554/eLife.39775.021

Additional evidence frequently invoked to support concerted channel gating and that we also found for SthK are the lack of subconductance states observed at the single-channel level and the equal single-channel amplitudes when the channel opens in response to cAMP or cGMP-binding (*Figure 1E*, *Figure 4B* and *Figure 4—figure supplement 2*). This suggests that there is only one structurally distinct open state, with the caveats that we are not able to observe states shorter than the dead time of the recording (~180 µs) and that structurally distinct open channel conformations may display the same single-channel current amplitude. Furthermore, in addition to the single 'resting' CNBD conformation reported here for SthK, there is also only one 'activated' CNBD conformation reported by the crystal structures of the isolated CNBDs of SthK (where the cAMP-bound is identical to the cGMP-bound conformation (*Kesters et al., 2015*), despite their quite different phenotypes at the functional level).

In contrast to what we observe with SthK, the CNBDs in the full-length apo HCN1 channel appear to be almost identical to those in the cAMP-bound HCN1, which are in the same 'activated' conformation seen in other CNBDs (*Lee and MacKinnon, 2017*) (*Figure 4—figure supplement 2*). Thus, in HCN1, cAMP binding does not lead to a drastic conformational change likely because the CNBDs are already 'pre-activated' in the absence of ligand. This is in agreement with functional data showing not only that cAMP binding has minimal effect on HCN1 channel activity but also that they are the fastest activating HCN channels, requiring the least amount of hyperpolarization for opening (*Santoro et al., 1998*).

## Model for agonist-induced channel activation

We do not have the structure of the full-length open SthK channel. However, based on the near identity structure-wise between the 'activated' CNBD structures of SthK (*Kesters et al., 2015*) and TAX-4 (*Li et al., 2017*) (*Figure 6—figure supplement 1*), we will use the full-length structure of TAX-4, the only open cGMP-bound CNG structure, as a proxy for open SthK. Thus, an overlay of SthK and TAX-4 structures reveal the regions in the channel that likely have to rearrange and how, in order to go from a closed to an open conformation. In TAX-4, the CNBD/C-linker domains are closer to the membrane, and are rotated outwards, ~9 degrees counterclockwise viewed from the extracellular side compared to SthK apo (*Figure 8C–D*). The B- and C-helices are shifted together as a rigid body to bring the C-helix in close contact with the ligand, and consequently the siphon is also displaced vertically towards the membrane to make room (Labeled as 1 and 2 in *Figure 8A*, *Video 1*). The C-linker from the adjacent subunit (labeled as 3 in *Figure 8A*) appears to have undergone a translation and rotation to accommodate for the displaced siphon, and since it is directly connected to the end of the S6 helices, they are consequently widened at the intracellular side. The resulting enlargement of the intracellular entryway likely has to be propagated to the selectivity filter to open the channel (*Figure 8A,C–D*).

Movements of the C-linker can also be transmitted to the S4-S5 linker of the adjacent subunit due to direct interactions between these regions (dashed circle *Figure 8A,B*). Displacements of the S5 transmembrane helix can be directly sensed by the pore helix and selectivity filter, a putative gate for these channels (*Flynn and Zagotta, 2001*), and bias its opening (*Mazzolini et al., 2018*). Additionally, this direct interaction between the S4-S5 linker and the CNBD also provides a direct link between cAMP binding and voltage sensor moving and may explain channel modulation by voltage.

## Conclusions

We report here the cryo-EM structures of apo, cAMP-bound and cGMP-bound SthK potassium channels in lipid nanodiscs. The conformations are identical, and functionally correspond to resting/closed states as indicated by single-channel recordings of SthK. Identical apo, and partial agonist-bound conformations indicate that SthK gating can be described with a concerted model, with strong coupling between the ligand-binding and pore domains. We hypothesize that the strong discrimination against cGMP in SthK is based on its preferred binding configuration to the apo conformation which differs from the configuration that leads to channel activation.

# Materials and methods

**Key resources table**

| Reagent type (species) or resource | Designation | Source or reference | Identifiers | Additional information |
|---|---|---|---|---|
| Gene (*Spirochaeta thermophila*) | SthK | NA | UniProtKB G0GA88 | |
| Cell line (*Escherichia coli*) | C41 (DE3) | Lucigen | 60442–1 | |
| Peptide, recombinant protein | SthK | doi: 10.1085/jgp.201812023 | | |
| Chemical compound, drug | cAMP | VWR | AAJ62174-03 | |
| Chemical compound, drug | cGMP | Sigma | G6129-100MG | |
| Software, algorithm | Relion 2.1 | doi: 10.7554/eLife.18722 | | |
| Software, algorithm | MotionCor2 | doi: 10.1038/nmeth.4193 | | |
| Software, algorithm | CTFFind4 | doi: 10.1016/j.jsb.2015.08.008 | | |

## Protein expression and purification

The C-terminally truncated gene for SthK 1–420 (UniProtKB G0GA88) was cloned into pCGFP-BC using restriction sites HindIII and XhoI (*Kawate and Gouaux, 2006*), which adds 19 additional amino acids from the multiple cloning site at the N-terminus as described previously (*Schmidpeter et al., 2018*). These 19 amino acids were not removed because they helped protein expression. The GFP and four out of the eight histidines were removed by Quikchange mutagenesis keeping the thrombin cleavage site. The construct with four histidines instead of eight displayed increased expression and solubility.

SthK was expressed in *E.coli* C41 (DE3) cells (Lucigen). Cells were grown in 4 L of LB media supplemented with 100 mg/L ampicillin at 37°C until reaching $OD_{600nm}$=0.4 and then transferred to 20°C. Protein expression was induced at $OD_{600nm}$=0.8 with 0.5 mM IPTG for 12 hr. The cells were harvested by centrifugation (5660 x g, 15 min, 4°C) and resuspended in 50 mL ice-cold lysis buffer (20 mM HEPES, pH 8.0, 100 mM KCl) supplemented with PMSF (85 µg/mL), Leupeptine/Pepstatin (0.95/1.4 µg/mL), 1 mg of DNaseI (Sigma), 1 mg of Lysozyme (Sigma), and one cOmplete ULTRA mini Protease Inhibitor (Roche) tablet. All following steps were performed at 4°C unless otherwise stated. The cells were lysed by sonication (Sonic Dismembrator 500, Fisher scientific) and solubilized with 30 mM n-Dodecyl-β-D-Maltopyranoside (DDM, Anatrace) for 1.5 hr. Non-solubilized material was removed by centrifugation (36,500 x g, 45 min, 4°C). The supernatant was supplemented with 40 mM imidazole, filtered through a 0.22 µm filter and applied to a 5 mL HiTrap chelating HP $Co^{2+}$ column (GE Lifesciences) equilibrated in wash buffer (20 mM HEPES, pH 8.0, 100 mM KCl, 40 mM imidazole, 200 µM cAMP, 1 mM DDM). The column was washed with 10–15 column volumes of wash buffer until a stable baseline was reached before eluting the protein with elution buffer (wash buffer supplemented with 250 mM imidazole). The protein was concentrated to 10 mg/mL using a 100 kDa cut-off concentrator (Amicon Ultra, Millipore) and applied to a Superdex 200 10/300 GL column (GE Lifesciences) equilibrated in running buffer (20 mM HEPES, pH 8.0, 100 mM KCl, 200 µM cAMP, 0.3 mM DDM) at room temperature. The peak fraction at ~11 mL containing SthK tetramer was collected and concentrated to 10–12 mg/mL (200–230 µM monomer) using a 100 kDa cut-off concentrator (Amicon Ultra, Millipore). The final protein concentration was determined using the molar extinction coefficient at 280 nm: 55,900 $M^{-1}cm^{-1}$ (ProtParam [*Wilkins et al., 1999*]).

## Nanodisc reconstitution

The plasmid of the membrane scaffold protein 1E3 (MSP1E3) was obtained from Addgene (#20064). Expression and purification of MSP1E3 was carried out as previously described, with some modifications (*Ritchie et al., 2009*): BL21(DE3) cells were used for protein expression at 37°C in 4 L LB media supplemented with 50 mg/L kanamycin. Expression was induced at $OD_{600nm}$=0.8 by 1 mM IPTG for 3 hr. Cells were harvested by centrifugation (5660 x g, 15 min, 4°C) and resuspended in 50 mL ice cold lysis buffer (40 mM Tris/HCl ($pH^{RT}$ 8.0), 300 mM NaCl, 1% Triton X-100) supplemented with PMSF (85 µg/mL), Leupeptine/Pepstatin (0.95/1.4 µg/mL), 1 mg of DNaseI (Sigma), 1 mg of Lysozyme (Sigma), and one cOmplete ULTRA mini Protease Inhibitor (Roche) tablet. All following steps were carried out at 4°C. The cells were lysed by sonication (Sonic Dismembrator 500, Fisher

scientific) and cell debris was removed by centrifugation (36,500 x g, 45 min, 4°C). The clarified lysate was incubated with 3 mL Ni-NTA resin (Millipore) (equilibrated in 40 mM Tris/HCl (pH$^{RT}$ 8.0), 300 mM NaCl, 1% Triton X-100) for 45 min. The resin was transferred into an EasyPack column (Bio-rad) and washed with 10 column volumes for each following buffers: wash buffer I (40 mM Tris/HCl (pH$^{RT}$ 8.0), 300 mM NaCl, 1% Triton X-100), wash buffer II (40 mM Tris/HCl (pH$^{RT}$ 8.0), 300 mM NaCl, 20 mM imidazole, 50 mM sodium cholate), wash buffer III (40 mM Tris/HCl (pH$^{RT}$ 8.0), 300 mM NaCl, 50 mM imidazole). The protein was eluted with wash buffer III containing 400 mM imidazole, concentrated using a 30 kDa cut-off concentrator (Amicon Ultra, Millipore) and applied to a PD10 desalting column (GE Lifesciences) equilibrated with 50 mM Tris/HCl (pH$^{RT}$ 8.0), 150 mM KCl, 0.5 mM EDTA for buffer exchange. The final MSP1E3 sample was concentrated to ~250 µM and stored at −80°C. For the lipid stock, 1-palmitoyl-2-oleoyl-*sn*-glycero-3-phospho-(1'-*rac*-glycerol) (sodium salt) (POPG, Avanti) in chloroform was dried under constant nitrogen stream forming a thin lipid film. Residual chloroform was further removed under vacuum for 3 hr. The lipid film was dissolved in 20 mM HEPES, pH 8.0, 100 mM KCl, 33 mM CHAPS by sonication at a final concentration of 25 mg/mL (32.4 mM) POPG and stored at −80°C.

Purified SthK was mixed with MSP1E3 and POPG prepared as above, in a molar ratio 1:1:70 (SthK (monomer):MSP1E3:POPG), supplemented with 1 mM cAMP and incubated for 1 hr at 4°C. Reconstitution was started by detergent removal using bio-beads SM-2 (Bio Rad, 20 mg per 100 µl mixture). After 3 hr of gentle shaking at 4°C, bio-beads were removed and the same amount of bio-beads was added for an additional 12 hr of continuous shaking at 4°C. The supernatant was collected, filtered through a 0.22 µm Spin-X centrifugation tube filter (Costar) and applied to a Superose 6 Increase (GE Lifesciences) equilibrated in running buffer (20 mM HEPES, pH 8.0, 100 mM KCl, 200 µM cAMP) at room temperature. The peak fraction corresponding to SthK in lipid nanodiscs was collected and concentrated to 5–5.5 mg/mL (approximated as 1 Absorption unit = 1 mg/mL) using a 10 kDa cut-off concentrator (Amicon Ultra, Millipore). For the preparation of SthK-apo and SthK-cGMP the size exclusion chromatography was equilibrated in cAMP-free running buffer. After concentrating, the sample was dialyzed against the cAMP-free running buffer for 5 hr at 4°C.

## Grid preparation and EM data collection

A negative stain protocol was published before (*Ohi et al., 2004*) and was used with minor modifications: 3 µL of purified SthK in lipid nanodics (20 µg/mL) was incubated for 60 s on a glow discharged homemade carbon coated copper grid (Electron Microscopy Sciences, 400 mesh). The grid was washed on four drops ddH2O (50 µL) and stained twice using a 1.5% uranyl acetate solution (50 µL). Excess liquid was removed by blotting after each step. For the second staining step the grid was incubated for 20 s in staining solution before blotting. Finally, the grid was air-dried for 5 min.

The prepared grids were loaded in a Tecnai T12 (FEI) electron microscope operated at 120 kV acceleration voltage equipped with a F416 (TVIPS) camera (2048 × 2048 pixel). Micrographs were taken at 49000x magnification with a pixel size of 3.44 Å/pixel using Leginon (*Suloway et al., 2005*). Reference-free particle picking was done using DoG Picker (*Voss et al., 2009*) and extracted particles were 2D classified with Xmipp (*Scheres et al., 2005a*; *Scheres et al., 2005b*) both included in the Appion (*Lander et al., 2009*; *Voss et al., 2010*) processing package.

For cryo-EM grid preparation, 2 mM cAMP and 7 mM cGMP were added to purified nanodiscs of SthK-cAMP and SthK-apo, respectively, to obtain SthK-cAMP and SthK-cGMP, and incubated for 30 min before freezing. Immediately prior to freezing, all samples were spiked with 3 mM fluorinated Fos-choline 8 (Anatrace). 3.5 µL of SthK in lipids nanodiscs were applied to glow-discharged UltrAu-Foil R1.2/1.3 300-mesh gold grids (Quantifoil). The sample was incubated for 10 s and then plunge frozen in liquid ethane using a Vitrobot Mark IV (FEI) with a blot force of 0 and a blot time of 2 s, at 22°C and 100% humidity.

Automated data collection was done in Leginon. The SthK-cAMP datasets were collected on a 300 keV Titan Krios (FEI) equipped with a K2 summit direct electron detector (Gatan) at a calibrated pixel size of 1.07325 A/pixel and a nominal defocus of −1.0 to −2.2 µm. The total dose of 70 e$^-$/Å$^2$ was distributed over 50 frames (1.4 e$^-$/Å$^2$/frame) at a total exposure time of 10 s (200 ms/frame).

The SthK-apo and SthK-cGMP datasets were collected on a different Titan Krios equipped with both a Quantum GIF and a K2 detector (Gatan, slit width 20 eV) and a Cs corrector at a calibrated pixel size of 1.0961 A/pixel and a nominal defocus of −1.0 to −2.2 µm. A total dose of 52 e$^-$/Å$^2$ was accumulated over 40 frames (1.3 e$^-$/Å$^2$/frame) and a total exposure time of 8 s (200 ms/frame).

## EM data analysis

The frame stacks were motion corrected using MotionCor2 (*Zheng et al., 2017*). CTFFIND4 (*Rohou and Grigorieff, 2015*) was used to determine the contrast transfer functions (CTF) on non-dose weighted summed images. All following image processing steps were carried out using the dose weighed summed images. Dogpicker (*Voss et al., 2009*) as part of the Appion processing package (*Lander et al., 2009*) was used for reference-free particle picking.

All processing steps were done with RELION 2.1 (*Scheres, 2012a, 2012b*; *Kimanius et al., 2016*) unless indicated otherwise. Three datasets for SthK-cAMP containing 5473 micrographs (823, 2210, 2440 for each dataset, respectively) were first separately subjected to two rounds of 2D classification using two-times binned particles. 572844 particles from all three datasets were combined after the 2D classification and further classified into 20 classes in 3D without applying symmetry using a 40 Å low-pass filtered map of HCN1 generated in Xmipp (*Sorzano et al., 2004*) as an initial reference (PDB: 5U6O). 233053 particles from four 3D classes showing a tetrameric channel arrangement were re-extracted, unbinned, and subjected to 3D Refinement applying C4 symmetry resulting in a 3.43 Å resolution map. From this point on, all processing steps were done with C4 symmetry applied. Further improvement of the resolution was achieved by focused classification using a mask excluding the nanodisc. The highest resolution class out of ten with most detailed features containing 81501 particles was subjected to a final 3D refinement. After conversion, the refinement was restarted applying a nanodisc-excluding mask. The resolution of refined maps was assessed by RELION postprocessing and the gold standard FSC value 0.143 using a mask that excluded the nanodisc. This resulted in a final resolution of 3.35 Å (*Figure 1—figure supplement 3*, EMDB-7484). In addition, the final particle set was subjected to the same refinement approach but without applying symmetry. The final resolution of the C1 refinement was 3.66 Å (*Figure 7B*).

A similar processing approach was used for SthK-cGMP combining two sets of data containing 2744 curated micrographs (1404, 1340, respectively) resulting in combined 199965 particles after individual 2D classification. 145635 selected particles after 3D classification (five out of ten classes) were refined to 3.60 Å. After focused classification 91800 particles (two out of five classes) were selected for 3D refinement with application of a mask excluding the nanodisc yielding a final map at 3.46 Å resolution with C4 symmetry (*Figure 1—figure supplement 4*, EMDB-7483) and 3.84 Å resolution without symmetry (*Figure 7C*).

SthK-apo structure was obtained from a single data collection session using the procedure described above. From 2130 micrographs 179504 particles were selected after 2D classification and subjected to 3D classification. The selected 121101 particles from the 3D classification (two out of five classes) resulted in a 3.79 Å resolution refined map. Focused classification resulted in 51115 particles (one out of five classes) refining to final resolution of 3.42 Å in C4 (*Figure 1—figure supplement 2*, EMDB-7482) and 3.90 Å in C1 (*Figure 7A*).

Local resolution distributions were calculated with blocres (*Cardone et al., 2013*) using the two half maps and the mask applied in postprocessing.

## Model building

For atomic model building a semi-de novo approach was used. One subunit of SthK-CNBD-cAMP crystal structure (PDB 4D7T) was docked into the density map of SthK-cAMP using UCSF Chimera (*Pettersen et al., 2004*). After a first round of refinement using phenix.real_space_refine (*Afonine et al., 2013*) miss-aligned regions were manually adjusted in COOT (*Emsley et al., 2010*). The transmembrane domain of one SthK subunit was then built de novo in COOT using bulky side chains (Phe, Trp, Arg) to establish the register of the alpha helices. The finished model of one SthK-cAMP subunit was subjected to real space refinement in PHENIX (*Afonine et al., 2013*). The tetramer was obtained by placing monomers into all four subunits (UCSF Chimera). The new model was refined in real space applying non-crystallographic symmetry and secondary structure restraints (PHENIX [*Adams et al., 2010*]). The atomic model of an individual SthK-cAMP subunit was used as initial model for both the SthK-cGMP and the SthK-apo structures and refined against the respective density maps, following the same process. Outliers were fixed manually, and specific regions in the protein that display very low resolution and lack of definition (the S2-S3 loop, and the last ~2 helical turns of the C-terminal helix) were removed using COOT in all final models. The side chains of several residues were not 100% defined, but since they were within a recognizable stretch with a clear

secondary structure and register, we decided to model them as the amino acid indicated in the sequence. The models were analyzed with Molprobity (*Chen et al., 2010*) and EMRinger (*Barad et al., 2015*) for validation. For further cross validation and to check for overfitting, all atoms of each model were randomly displaced by 0.5 Å and each resulting model was refined against the first half-map obtained from processing. We then calculated the FSC between the refined models and the half-maps used during refinement and compared them with the FSC between the refined models and the other half-maps, not used during refinement. In addition, the FSC between the refined model and sum of both half maps was calculated. The nanodisc area in all maps was excluded by applying the same mask used in RELION postprocessing to prevent non-meaningful correlation at low resolution. Correlations at higher resolution were not influenced by the mask. The resulting FSC curves were similar showing no evidence of overfitting. The final models contained amino acids 10–62 and 76–412 (SthK-cAMP, PDB 6CJU), 10–68 and 76–412 (SthK-cGMP, PDB 6CJT), and 10–65 and 80–415 (SthK-apo, PDB 6CJQ). HOLE (*Smart et al., 1996*) was used to measure the pore size and water accessible cavities in the voltage sensor domain. 'Omit' maps between experimental data and atomic models were calculated using phenix.real_space_diff_map (*Afonine, 2017*) (PHENIX). Difference maps between two experimental maps were obtained using the program diffmap from the Grigorieff laboratory (http://grigoriefflab.janelia.org/diffmap). Figures were prepared in Chimera (*Pettersen et al., 2004*), VMD (*Humphrey et al., 1996*) and PyMOL Molecular Graphics System, Version1.8 Schrodinger, LLC.

## Liposome reconstitution

Chloroform-solubilized lipids: 1,2-dioleoyl-*sn*-glycero-3-phosphocholine (DOPC), POPG and 18:1 Cardiolipin were mixed in a glass tube at a ratio 5:3:2 (w:w:w), respectively. Lipids were dried under nitrogen, washed with pentane, and redried. The thin lipid film was solubilized immediately by sonication in reconstitution buffer (10 mM HEPES, pH 7.6, 400 mM KCl, 5 mM NMDG) containing 33 mM CHAPS, at a concentration of 10 mg lipids/mL. Solubilized lipids were mixed with freshly purified cAMP-bound SthK (30 µg protein/mg lipid). Detergent was removed by applying 0.5 mL of the mixture to an 18 mL self-packed, degassed, and pre-swollen in reconstitution buffer Sephadex G-50 (GE Lifesciences) column. Liposomal aliquots (eluting at about 7 mL) are collected in 0.5 mL fractions, visually inspected for turbidity and frozen and stored in small aliquots at −80°C.

## Single-channel recording and analysis

Electrophysiological experiments were performed using a horizontal lipid bilayer setup as previously described (*Heginbotham et al., 1999*; *Posson et al., 2013*). Briefly, an artificial lipid bilayer was formed over a ~100 µm diameter hole in the partition separating the *cis* (top) and *trans* (bottom) chamber of the recording system by painting a 6.25 mg/mL solution of 1,2-dioleoyl-*sn*-glycero-3-phosphocholine (DPhPC) in *n*-decane (Sigma). The solutions in both chambers contain 97 mM KCl, 3 mM KOH, 10 mM Hepes, pH 7.0 . Agar bridges in the two chambers connect via Ag/AgCl pellet electrodes to an Axopatch 200A/B amplifier (Molecular Devices). All electrophysiological signals are filtered at 1000 Hz with an 8-pole low-pass Bessel filter, digitized at 20 KHz with Digidata 1440A (Molecular Devices) and collected using Clampex software (V10, Molecular Devices). Bilayer formation was monitored electronically using an in-house designed protocol in Clampex. Freshly-thawed and briefly sonicated SthK-containing liposomes were applied to the *cis*-side of the bilayer, and fusion events were monitored in the Gap-Free recording mode in Clampex under +100 mV applied voltage. The channel orientation was determined by having cAMP only on one side of the bilayer, since SthK displays no activity in the absence of cAMP. The system was set up so that only channels with the CNBDs facing the *trans* chamber were recorded. The *trans* chamber was serially perfused to different conditions to determine the channel response to cyclic nucleotides. The single-channel data was analyzed in Clampfit (Molecular Devices) with no additional filter applied unless stated otherwise. The dwell-time analysis was performed using the Single-Channel Search module in Clampfit. The dwell-time histograms include only events longer than 360 µs (two times the dead time). The closed dwell-time histograms additionally do not contain the very rare events longer than 500 ms. All plots and fits were made in Clampfit and then imported into Adobe Illustrator for final figure assembly. The single-channel measurements were repeated for at least 7 separate bilayers for each condition reported in *Figure 1* and *Figure 4* and yielded similar results. The all-amplitude

histograms and dwell-time distributions for all the single-channel data obtained are qualitatively similar with the representative ones shown in *Figure 4—figure supplement 2*.

## Acknowledgements

All EM data collection and screening was performed at the Simons Electron Microscopy Center and National Resource for Automated Molecular Microscopy located at the New York Structural Biology Center, supported by grants from the Simons Foundation (349247), NYSTAR, and the NIH National Institute of General Medical Sciences (GM103310). We would like to specifically thank L Kim, A Raczkowski and K Jordan for their technical support during grid screening and L Kim, W Rice and E Eng for their support during data collection. Grid preparation and initial screening were performed at the Electron Microscopy Resource Center at Rockefeller University with the help of D Acehan. Initial negative stain screening was performed at the Weill Cornell Microscopy and Image Analysis Core Facility, with the help of L Cohen-Gould. We would like to thank H Stahlberg for introduction to cryo-EM, V Moiseenkova-Bell for helpful hints for grid preparation, I Kiburu and M Falzone for helpful discussions. We thank S Scheuring and A Accardi for critically reading the manuscript. This work was supported in part by National Institutes of Health grants (R01GM124451 and R01GM088352) to CMN and the German Research Foundation DFG (SCHM 3198/1–1) to PS. Accession numbers of the apo, cAMP- and cGMP-bound structures of the prokaryotic SthK channel are: 6CJQ, 6CJU and 6CJT (coordinates of atomic models), EMD-7482, EMD-7484 and EMD-7483 (density maps).

## Additional information

### Funding

| Funder | Grant reference number | Author |
|---|---|---|
| Deutsche Forschungsgemeinschaft | SCHM 3198/1-1 | Philipp AM Schmidpeter |
| National Institute of General Medical Sciences | R01GM124451 | Crina M Nimigean |
| National Institute of General Medical Sciences | R01GM088352 | Crina M Nimigean |

The funders had no role in study design, data collection and interpretation, or the decision to submit the work for publication.

### Author contributions

Jan Rheinberger, Conceptualization, Resources, Data curation, Software, Formal analysis, Validation, Investigation, Visualization, Methodology, Writing—original draft, Writing—review and editing, Expresses and purified protein, Prepared cryo-EM samples, Collected and processed cryo-EM data, Built and refined atomic model; Xiaolong Gao, Formal analysis, Validation, Investigation, Writing—review and editing, Performed electrophysiology experiments; Philipp AM Schmidpeter, Resources, Validation, Investigation, Writing—review and editing, Performed initial protein purification and electrophysiology characterization; Crina M Nimigean, Conceptualization, Resources, Data curation, Formal analysis, Supervision, Funding acquisition, Validation, Investigation, Visualization, Methodology, Writing—original draft, Project administration, Writing—review and editing

### Author ORCIDs

Jan Rheinberger (iD) http://orcid.org/0000-0002-9901-2065
Xiaolong Gao (iD) http://orcid.org/0000-0001-8933-9286
Philipp AM Schmidpeter (iD) https://orcid.org/0000-0003-2871-9706
Crina M Nimigean (iD) http://orcid.org/0000-0002-6254-4447

### Decision letter and Author response

Decision letter https://doi.org/10.7554/eLife.39775.037
Author response https://doi.org/10.7554/eLife.39775.038

## Additional files

### Supplementary files
• Supplementary file 1. Table Summary of imaging conditions, data processing and model statistics.
DOI: https://doi.org/10.7554/eLife.39775.022
• Transparent reporting form
DOI: https://doi.org/10.7554/eLife.39775.023

### Data availability
The 3 density maps and 3 atomic models have been deposited in PDB under the following accession codes: 6CJQ, 6CJU and 6CJT (coordinates of atomic models), EMD-7482, EMD-7484 and EMD-7483 (density maps).

The following datasets were generated:

| Author(s) | Year | Dataset title | Dataset URL | Database, license, and accessibility information |
|---|---|---|---|---|
| Nimigean CM, Rheinberger J, Gao X, Schmidpeter PAM | 2018 | apo-bound structure of the prokaryotic SthK channel (coordinates of atomic model) | https://www.ebi.ac.uk/pdbe/entry/pdb/6cjq | Publicly available at the Electron Microscopy Data Bank (accession no: 6CJQ) |
| Nimigean CM, Rheinberger J, Gao X, Schmidpeter PAM | 2018 | cAMP-bound structure of the prokaryotic SthK channel (coordinates of atomic model) | https://www.ebi.ac.uk/pdbe/entry/pdb/6cju | Publicly available at the Electron Microscopy Data Bank (accession no: 6CJU) |
| Nimigean CM, Rheinberger J, Gao X, Schmidpeter PAM | 2018 | cGMP-bound structure of the prokaryotic SthK channel (coordinates of atomic model) | https://www.ebi.ac.uk/pdbe/entry/pdb/6cjq | Publicly available at the Electron Microscopy Data Bank (accession no: 6CJT) |
| Nimigean CM, Rheinberger J, Gao X, Schmidpeter PAM | 2018 | apo-bound structure of the prokaryotic SthK channel (density map) | http://www.ebi.ac.uk/pdbe/entry/emdb/EMD-7482 | Publicly available at the Electron Microscopy Data Bank (accession no: EMD-7482) |
| Nimigean CM, Rheinberger J, Gao X, Schmidpeter PAM | 2018 | cAMP-bound structure of the prokaryotic SthK channel (density map) | http://www.ebi.ac.uk/pdbe/entry/emdb/EMD-7484 | Publicly available at the Electron Microscopy Data Bank (accession no: EMD-7484) |
| Nimigean CM, Rheinberger J, Gao X, Schmidpeter PAM | 2018 | cGMP-bound structure of the prokaryotic SthK channel (density map) | http://www.ebi.ac.uk/pdbe/entry/emdb/EMD-7483 | Publicly available at the Electron Microscopy Data Bank (accession no: EMD-7483) |

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
