## [Decision Letter]

Thank you for submitting your article "Ligand discrimination and gating in CNG channels from apo and partial agonist-bound cryo-EM structures" for consideration by *eLife*. Your article has been reviewed by Richard Aldrich as the Senior Editor, Reviewing Editor and Reviewer #1. The Reviewing Editor has drafted this decision to help you prepare a revised submission.

Summary:

This is an interesting manuscript reporting three new cryo-EM structures of the prokaryotic SthK channel from *Spirochaeta thermophila*. SthK is a K^+^ selective channel that is modulated by cyclic nucleotides binding to the cyclic nucleotide binding domain (CNBD) in the intracellular C-terminus of the channel. Single channel data presented here shows that SthK is activated by cAMP and by membrane depolarization, although the open probability of the channel at 0 mV is only 0.15 in the presence of what appears to be a saturating concentration of cAMP, making this nucleotide a relatively weak agonist. Only occasional openings are observed in the presence of cGMP, making this nucleotide an even weaker agonist. The authors solve the structure of SthK in the absence of nucleotides (apo) and in the presence of either cAMP or cGMP. This is the first structure of a full-length cyclic nucleotide regulated channel in the apo state. One of the most interesting features of the structures is that all three are very similar, the pore appears closed and nucleotide binding does not cause any detectable structural change even in the CNBD. The authors clearly discuss the implications of their work for understanding the mechanisms of gating in the larger family, and interpret their structures as support for an allosteric mechanism of gating. The manuscript received thoughtful review at another journal, and the authors have revised the manuscript to address each of the points raised. The reviews and revised manuscript were reviewed by the Editor, who decided to invite revision to address several remaining issues that can be addressed with careful revision.

Essential revisions:

1) The discussion in the subsection “Ion channel pore” concerning the location of the gate seems counterintuitive. The diameter of the pore at I215 in SthK would seem to suggest this relatively intracellular region would be a barrier to ion permeation of hydrated ions. The most straightforward interpretation of the structure would be that the intracellular pore is likely a gate and that the selectivity filter may or may not play an additional role. The authors should consider revising this section.

2) In the subsections “SthK architecture” and “Voltage sensor”, when discussing the S1-S4 domains, the authors should tone down assertions that this domain is responsible for the voltage-dependent behavior they observe. That domain may serve as a voltage sensor in SthK, but at this point it is speculation and should be stated to be such. There are now a lot of structures of ion channels with S1-S4 domains (CNG, TRP) that have yet to implicated in voltage sensing.

3) Please go through all legends and make sure the concentration of cyclic nucleotides is specified. Figure 2E and Figure 4—figure supplement 1 are good examples.

4) The authors have no constraint on how many figures can appear in the main text, and the authors should consider moving some out of the supplement. Supplementary figure 9E for example, should clearly be moved to appear alongside Figure 2E. The other parts of Supplementary figure 9, Supplementary figure 7 and Supplementary figure 11 might be better to appear in the main text.

---

## [Author Response]

Essential revisions:1) The discussion in the subsection “Ion channel pore” concerning the location of the gate seems counterintuitive. The diameter of the pore at I215 in SthK would seem to suggest this relatively intracellular region would be a barrier to ion permeation of hydrated ions. The most straightforward interpretation of the structure would be that the intracellular pore is likely a gate and that the selectivity filter may or may not play an additional role. The authors should consider revising this section.

The section was revised according to the reviewer’s suggestion. The new text (in the subsection “Ion channel pore”) now mentions residue I215 as possible location of the gate with a potential additional gate at the selectivity filter.

2) In the subsections “SthK architecture” and “Voltage sensor” when discussing the S1-S4 domains, the authors should tone down assertions that this domain is responsible for the voltage-dependent behavior they observe. That domain may serve as a voltage sensor in SthK, but at this point it is speculation and should be stated to be such. There are now a lot of structures of ion channels with S1-S4 domains (CNG, TRP) that have yet to implicated in voltage sensing.

We agree that at the current point the assignment of the S1-S4 domain as voltage sensor was an educated guess based on comparison with similar channels, and that further experiments would be necessary for validation. We are now stating that the S1-S4 form a voltage sensor-like domain (subsection “SthK is a cAMP-gated potassium channel”, last paragraph) and toned down our statements about the function of this domain (subsection “Ion channel pore”).

3) Please go through all legends and make sure the concentration of cyclic nucleotides is specified. Figure 2E and Figure 4—figure supplement 1 are good examples.

We update the following figure legends and specified the cyclic nucleotide concentrations: Figure 4B, Figure 4C (formerly known as Figure 2E and Supplementary figure 9C), Figure 1—figure supplement 3, Figure 1—figure supplement 4, Figure 4—figure supplement 2.

4) The authors have no constraint on how many figures can appear in the main text, and the authors should consider moving some out of the supplement. Supplementary figure 9E for example, should clearly be moved to appear alongside Figure 2E. The other parts of Supplementary figure 9, Supplementary figure 7 and Supplementary figure 11 might be better to appear in the main text.

We heeded the reviewers’ suggestion. We now have only 10 supplementary figures instead of 13, and many panels from the remaining figures have been transferred to the main text. We moved former Supplementary figure 9C into now Figure 4C. We moved panel A-B in former Supplementary figure 9 into Figure 4A replacing the previous inserts. We moved former Supplementary figure 7 into Figure 2. We moved former Supplementary figure 11B into Figure 6F, and former Supplementary figure 13 is now Figure 7.

In addition, we split former Figure 2 into two figures (Figures 3 and 4), one dedicated to the pore, and the other to the voltage sensor.